# Comparing Public and Provider Preferences for Setting Healthcare Priorities: Evidence from Kuwait

**DOI:** 10.3390/healthcare9050552

**Published:** 2021-05-08

**Authors:** Abdullah M. Alsabah, Hassan Haghparast-Bidgoli, Jolene Skordis

**Affiliations:** 1Institute for Global Health, University College London, London WC1N 1EH, UK; h.haghparast-bidgoli@ucl.ac.uk (H.H.-B.); j.skordis@ucl.ac.uk (J.S.); 2Medical Services Authority, Ministry of Defence, Kuwait City 13012, Kuwait

**Keywords:** public preferences, service provider preferences, priority setting, resource allocation, Kuwait

## Abstract

As attempts are made to allocate health resources more efficiently, understanding the acceptability of these changes is essential. This study aims to compare the priorities of the public with those of health service providers in Kuwait. It also aims to compare the perceptions of both groups regarding key health policies in the country. Members of the general public and a sample of health service providers, including physicians, dentists, nurses, and technicians, were randomly selected to complete a structured, self-administered questionnaire. They were asked to rank health services by their perceived importance, rank preferred sources of additional health funding, and share their perceptions of the current allocation of health resources, including current healthcare spending choices and the adequacy of total resources allocated to healthcare. They were also asked for their perception of the current local policies on sending patients abroad for certain types of treatments and the policy of providing private health insurance for retirees. The response rate was above 75% for both groups. A higher tax on cigarettes was preferred by 73% of service providers as a source of additional funding for healthcare services, while 59% of the general public group chose the same option. When asked about the sufficiency of public sector health funding, 26.5% of the general public thought that resources were sufficient to meet all healthcare needs, compared with 40% of service providers. The belief that the public should be offered more opportunities to influence health resource allocation was held by 56% of the general public and 75% of service providers. More than half of the respondents from both groups believed that the policy on sending patients abroad was expensive, misused, and politically driven. Almost 64% of the general public stated that the provision of private health insurance for retirees was a ‘good’ policy, while only 34% of service providers agreed with this statement. This study showed similarities and differences between the general public and health service providers’ preferences. Both groups showed a preference for treating the young rather than the old. The general public preferred more expensive health services that had immediate effects rather than health promotion activities with delayed benefits and health services for the elderly. These findings suggest that the general public may not accept common allocative efficiency improvements in public health spending unless the challenges in this sector and the gains from reallocation are clearly communicated.

## 1. Introduction

Rising non-communicable disease (NCD) prevalence and falling oil revenues pose an unprecedented health financing challenge in Kuwait [1], a context where public health expenditure accounted for 84% of total health spending in 2016 [2]. The Kuwaiti government may no longer have sufficient funds to finance all previously funded health services and may need to engage in an explicit priority setting process to improve the efficiency of the system.

Most health services are provided in the country, but in cases where treatment is complicated or not available locally, the government sends patients overseas for treatment. The cost of this service comes at a substantial expense to the government and has been increasing over the past few years, making it particularly controversial given the country’s current economic situation. This service is exclusive to Kuwaiti nationals. In these cases, the government does not only pay for the treatment of patients, but also pays patients and their companions for living allowances and flight tickets. The latest policy stated that a patient gets paid 75 KD per day (around $249), 50 KD per day (around $166) for the first chaperone, and only flight tickets for the second chaperone [3]. Table 1 shows government spending on sending patients abroad for treatment in three consecutive fiscal years.

Another initiative health authorities in the country took was to improve the overall quality of services by increasing the private sector’s involvement in health provision [4]. One of the first steps that these authorities took was the procurement of private health insurance on behalf of retirees. This policy was issued by the parliament in 2014, the contract was signed between the Ministry of Health and the insurance company that won the bid in July 2016, and the provision of services for beneficiaries started in October 2016 [5]. The contract value was around $272.7 million for the first year, with a cost of around $2549 per person [5]. Initially, the Ministry of Health expected the number of beneficiaries who utilised services to be 107,000 during the first year of the contract; however, the actual number of retirees who utilised services was 114,952, and is expected to increase to 125,000 by the second year of the contract [5]. The service network is comprised of 120 local health practices and more than 800 doctors, providing inpatient services, chronic and specialised outpatient services, dental services, obstetric services, and others.

Making choices to allocate resources between competing services is known as priority setting [6,7,8,9,10,11]. It is a process that includes different disciplines such as ethics, economics, politics, and epidemiology [8,10,12,13,14,15,16]. Therefore, at all levels of a health system, the process of priority setting is believed to be continuous, complex, and challenging [17]. Traditionally, policymakers and hospital managers have performed the process of priority setting and resource allocation within a health system [13,17,18,19].

Despite the argument that there is no single clear process for setting priorities in a health system [18], there have been several attempts to identify frameworks that might improve or expedite the priority setting [9,17]. There are increasing demands to increase the public’s involvement in health and healthcare decision-making [20,21]. It has been further argued that there is a need to include the public’s preferences on healthcare priority setting to inform the decision-making process for it to be considered legitimate and acceptable [21,22]. Several studies have evaluated various methods of involving the general public in setting healthcare priorities [23,24,25,26,27,28]. Such initiatives have been performed in high- [22,29,30,31,32,33,34], middle-, and low-income countries [35].

Despite some progress, a remaining challenge for decision-makers is how to involve members of the public in setting healthcare priorities and allocating health resources in a range of health systems [23,36]. It is also important to identify who should be accountable for making decisions regarding resource allocation in public healthcare since there is arguably a gap between public expectations and healthcare resources [32]. Additionally, there are concerns about establishing a ‘dictatorship of the uninformed’ by those more cautious of the idea of involving the public in setting healthcare priorities [34].

This study aims to compare the healthcare priorities of the general public with those of health service providers in Kuwait. Such priorities include how important both groups believe some health services are, their preferred sources for health funding, their attitudes towards the current allocation of health resources, the perceptions of current healthcare costs, and the adequacy of currently available resources. It also aims to compare both groups’ perceptions regarding two key health policies in the country, namely sending patients abroad for certain types of care and providing private health insurance for retirees. Findings from this study will help provide decision-makers with useful information to develop and implement informed priority setting strategies in the Kuwaiti health sector with useful information to develop more informed priority setting strategies.

## 2. Materials and Methods

### 2.1. Study Population and Sampling

This study includes a sample of health service providers and the general public in Kuwait. The sample representing the general public was randomly selected to participate in the survey. The sample included residents of Kuwait (both Kuwaitis and non-Kuwaitis) aged 21 and older who were currently living in Kuwait. This sample excluded non-Arabic or non-English speakers, temporary visitors to the country, or individuals who did not consent to participate in the study.

To calculate the survey sample size, a prevalence of a certain opinion of 50.0% was assumed, along with a 95% confidence interval (CI) (Z = 1.96), a 5% acceptable margin of error, a simple sampling design effect coefficient of 1, and two groups of comparison according to nationality (Kuwaitis and non-Kuwaitis). Calculations resulted in a sample size of 768.3 individuals, which was further increased by 50% (1537) to account for contingencies such as non-response and recording errors.

The following formula was used for sample size calculation: $$n={\left(Z\right)}^{2}\frac{p\left(1-p\right)}{e^{2}}$$
where:

*Z* = level of confidence measure. It describes the level of uncertainty in the sample prevalence as an estimate of the population prevalence. Recommended value: 1.96 (for a 95% confidence level).

*p* = baseline levels of the indicators. The estimated prevalence of a certain opinion within the target population. Values closest to 50% are the most conservative. Recommended value: 0.5 if no previous data on the population are available.

*e* = margin of errors. The expected half-width of the confidence interval. The smaller the margin of error, the larger the sample size needed. Recommended value: 0.05.

Expected response rate = the anticipated response rate. Recommended value for opinion research: 0.5 as an estimate if no previous data on the population for the general public, not for the care providers, are available.

A simple random sampling procedure was carried out to randomly select participants from the target population in public areas and workplaces. Although stratified, systematic, and even cluster sampling are ideal for nationwide surveys, the simple random sampling technique was the most suitable for this opinion research due to logistics and time limitations. Data collection took place in February 2018. Six days of each week were selected at random as follows: a 3-day period for the general public in malls and grocery stores starting at 6:00 p.m. for 6 h per day, and another 3-day-period for both the workplaces of the general public sample, starting at 1:30 p.m. for two hours per day, and for the service providers, starting at 7:30 a.m. for 4 h per day, with one day off per week.

Regarding the sample of the service providers, a comprehensive list of the target population was generated using the Health Manpower and Medical Licensing departments of the MOH. From this list, a simple random sample was selected to participate in the survey. This sample included different health service providers (physicians, dentists, nurses, pharmacists, and technicians) and excluded those on leave during data collection and those who did not consent. A simple random sampling procedure was carried out to randomly select participants from the target population of health service providers registered in the Health Manpower Directorate, MOH; the Kuwait Medical Association (KMA) database; and the Directorate of Medical Licensing. The sample included individuals working in either public or private healthcare facilities from all six governorates of Kuwait. Table 2 shows the distribution of personnel by category of provider and type of establishment according to the last quarter of 2017. Healthcare workers associated with “Oil” are those who work for the oil sector companies’ hospital (Table 2).

### 2.2. Data Collection

Data were collected from individuals using a structured questionnaire that included questions from similar previous studies [20,26,28,29,32,33,38], as well as questions that were developed through interviews with hospital managers as part of a qualitative study conducted by the authors from a previous paper [39]. The questions were formulated to be easily comprehensible for health service providers and laypeople alike. The questionnaire was translated from English to Arabic, since it was the main spoken language in the country, and then back-translated into English. Participants had the option of completing a questionnaire in their preferred language.

The first part of the questionnaire included questions about respondents’ sociodemographic characteristics, while the second part was intended to illustrate the care-seeking behaviour of respondents. Participants were then asked to rank 12 services and treatments according to their importance in the third part of the questionnaire. The services were adapted from previous studies and aimed to check participants’ preferences for the allocation of resources towards younger versus older patients. It also evaluated whether respondents preferred more expensive health services with immediate effects rather than prevention and/or health promotion services where the benefits may be delayed. Respondents were then asked about their preferred sources of additional funding for health services in the fourth part of the questionnaire. Next, respondents’ attitudes with respect to resource allocation in healthcare were evaluated in the fifth part, while their perceptions regarding healthcare costs were covered in the sixth part, and their perceptions on the adequacy of healthcare resources were assessed in the seventh part of the questionnaire. The eighth and ninth sections of the questionnaire asked participants about their perceptions regarding some ‘hot topics’ in the health policy arena in Kuwait, namely the policies on overseas treatment and private health insurance for retirees and the number of resources allocated for such schemes.

A team of six data collectors attended a three-day training workshop on the objectives of the study, the content of the questionnaire, expected questions from participants, the content and purpose of the information sheet shared with respondents, and the informed consent form. The primary researcher led the workshop from the 28th to the 30th of January 2018. The data collectors had experience in data collection from previous surveys performed in the country by the Ministry of Health and other international organisations. Data collectors were asked to explain the survey’s objectives and obtain written informed consent from participants before starting the survey.

After training, data collectors performed a pilot study that included 30 purposively selected individuals from the general public and service providers. This phase aimed to check if the questions were easy to understand and to evaluate the team’s data collection skills. The questionnaire was modified by changing and/or removing some questions that were hard to understand. Respondents spent 15–20 min completing the questionnaire during the pilot.

The study took place in all six governorates of Kuwait for the general public and health service providers. As for the general public, public areas in the centre of governorates with easy accessibility were chosen for data collection. Respondents were approached in shopping malls, grocery stores, governmental workplaces, private companies, and the Public Authority for Social Security. Permissions were acquired to interview respondents at the previously mentioned locations. In each of the locations, participants were recruited from their desks as well as building reception areas. A designated area was chosen for the interview to take place in each of the locations. As for service providers, the interviews were conducted in clinics and hospitals to have a representative sample of selected respondents.

This location was chosen because it is usually visited by retirees, who are chief targets of this study since they are beneficiaries of one of the healthcare policies intended to be evaluated (private health insurance for retirees).

Questionnaires were self-administered. Each questionnaire was given a unique code and was checked for completion.

### 2.3. Data Management and Analysis

Data collected was doubly entered. First, the data were entered by one data operator from the Kuwait National Centre for Health Information (NCHI). Then, the primary researcher entered the collected data using Microsoft Excel^®^. Double-checking was performed where any inconsistencies were corrected after confirmation from the hard copies. Each questionnaire had a unique code. Data analysis was performed using SPSS version 24, using appropriate methods for the sample design of the survey. Entered data were checked for accuracy, then for normality, using the Kolmogorov–Smirnov and Shapiro–Wilk tests, respectively.

The following statistical tests were used:Independent samples Mann–Whitney’s U-test (or Z-test) was used as a nonparametric test of significance for comparing two sample medians. This study used ordinal variables such as age groups, education, monthly income, a ranking of health services, and Likert scale questions. For negatively worded items, the median was calculated in the reverse direction; however, for tabular presentation, these were left as the respondent answered.The χ^2^-test (or Yate’s corrected chi-square test) was used as a non-parametric test of significance for comparing the distributions of two qualitative variables.

A 5% level was chosen as a level of significance in all statistical significance tests used.

Few participants chose to have ‘no response’ for some general characteristics, which did not exceed 5% of the corresponding sample. In the presentation, they were kept because of their minority and because no association with other questionnaire items was identified.

### 2.4. Ethical Approval

Ethical approval for the study was obtained from University College London (9633/001) and the Standing Committee for Coordination of Medical Research, MOH, Kuwait (Meeting number 5/2016). An information sheet was given to each participant explaining the study’s objectives and informed consent was obtained from each participant before proceeding with the survey.

## 3. Results

The results showed a response rate of 78.8% for the general public (n = 1211) and 75.2% for health service providers (n = 578). Table 3 shows the demographics of the respondents. 

In both groups, the majority had used a public healthcare facility for their last care-seeking visit. Table 4 shows that 63% of the general public and 30% of service providers had visited a healthcare facility in the last month. Regarding payment method, 34% of the general public had used public health services (free of charge), and 21% paid with out-of-pocket payment, while 43% of service providers used public services and approximately 29% paid with out-of-pocket payment.

The results showed that most of the variables were not normally distributed. In order to know the respondents’ priorities for health services, they were asked to rank their priorities according to their importance. Table 5 shows a comparison of the mean priority rankings for 12 services and treatments adapted from other studies [20,33] between the general public and service providers. The agreement of top priorities is remarkable, with both groups agreeing on the first and third priorities, and both groups agreeing on the top 4 priorities. The table shows that the three highest priorities for the general public were “treatment for children with life threatening illnesses”, followed by “special care and pain relief for people who are dying”, and “high technology surgery, organ transplants and procedures which treat life threatening conditions” came in as the third priority. However, service providers had “preventive screening services and immunization” as their second priority, while having the same first and third priorities as the general public. The general public believed that “nursing and community services at home” was their lowest priority, while providers thought that the extended stay hospital care for the elderly was of the least important priorities.

Respondents were asked, “in your opinion, what other sources do you prefer for additional funding for healthcare services in the country?”. The results of this question are shown in Table 6. The most popular option for additional funding for both groups was a higher tax on cigarettes, while the least popular option for both groups was implementing an income tax. 44.5% of service providers preferred to decrease the budget allocated for sending patients abroad as a source of additional funding, while only 19% of the general public chose the same option (*p* < 0.05). Other suggestions for additional funding included the more efficient management of public health services, a reduction in the number of high government officials, and decreasing the budgets allocated for hospitality and gifts in the governmental sector.

The study also focused on the attitudes of respondents with respect to resource allocation in healthcare. Table 7 illustrates the attitudes of the general public and service provider groups towards some resource allocation questions adapted from a previous study [26]. A majority in both groups agreed with the top five opinions (ranked in Table 7). Around 75% of service providers believed that the general public should be offered more opportunities to influence healthcare resource allocation, while only 56% of the general public agreed with this question. Regarding the respondents’ attitudes towards decision-makers in the Ministry of Health, only 13% of the general public and 32% of service providers thought that decision-makers were handling prioritisation in a good manner. Additionally, 80.3% of the general public and 57.1% of service providers believed that more explicit prioritisation should be made.

In order to understand participants’ perceptions regarding healthcare costs, they were asked to give their opinion on some statements adapted from a previous study [28]. Table 8 shows the general public and health service providers’ responses towards some statements related to healthcare costs. Around 78% of the general public and 86% of service providers chose to use a less effective but cheaper treatment if two types of treatment exist for a certain disease. More than half of service providers disagreed that money is spent on unnecessary healthcare costs, while only 40% of the general public disagreed with the same statement.

Respondents’ perceptions of the adequacy of healthcare resources were also evaluated. Table 9 illustrates the views of the respondents. No health service was perceived as “Too much”, indicating a general dissatisfaction for all services. More than half of respondents from both groups thought that dental services received enough resources. With hospital care services, more than half of the general public stated that the resources allocated were too few, while around half of service providers believed that the resources received by this service were adequate. Half of service providers thought that primary healthcare services received enough resources, but fewer from the general public believed that this service received less than adequate resources. Almost half of service providers thought that child care services were allocated enough resources, but 37.2% of the general public believed that these services received too few resources.

The perception of the general public and health service providers regarding some ‘hot topics’ in the country’s health policies and the number of resources allocated for such schemes were important to this study. Two of these policies were sending patients abroad for treatment and private health insurance for retirees (Afya). The following section included Kuwaiti participants’ responses only, since these policies are exclusive to Kuwaiti nationals.

A large majority from the general public is aware of the policy on sending patients abroad for treatment, and most of them, or one of their relatives, had benefited from this service. Surprisingly, fewer of the service providers knew of the policy. Table 10 summarises these findings.

The results showed some differences in both groups’ attitudes towards the policy on sending patients abroad for treatment that were statistically significant. More than half of the respondents from both groups believed that the policy was costly/expensive, misused, and politically driven. Members of the general public agreed more with these statements than the service providers. More than half of the general public believed that beneficiaries of this policy were sent abroad without real medical indication, whereas 42% of service providers agreed with this statement. Around 38% of service providers and 32% of the general public respondents expressed their acceptance that most specialised treatments were available in the country. When asked if this policy was a constitutional right for citizens, around 44% of the general public agreed with this statement, while around 30% of service providers agreed. When asked if this policy had advantages, more than half of service providers neither agreed nor disagreed, while around 65% of the general public stated that this policy did have several advantages. More than half of the participants from both groups agreed that this policy decreased the trust in the local healthcare system.

Regarding the policy on private health insurance for retirees (Afya), the results showed that most of the general public are aware of this policy, and expressed that they or their relatives benefited from this scheme. Similar to the policy of sending patients abroad, fewer service providers knew of this policy. Table 11 shows the general public and service providers’ responses to some statements, which were also mentioned by hospital managers in a previous qualitative study about the policy on purchasing private health insurance on behalf of retirees (Afya).

Similar to the respondents’ attitudes towards the policy on sending patients overseas for treatment, their attitudes towards the policy on private health insurance for retirees (Afya) also had significant differences. Almost 64% of the general public stated that Afya was a good policy, while only around 34% of service providers agreed with this statement. More than half of the general public believed that the policy had clear objectives and decreased the load on public health services. On the other hand, only 27% of service providers thought that the policy had clear objectives, and 44% of them expressed that the policy decreased the load on public health services. While the majority of the general public thought that Afya promoted patient choice for its beneficiaries, only 44% of service providers shared the same understanding. Around 48% of respondents from the general public and 36% of service providers did not think that the policy was a step towards privatising healthcare. Another difference in attitudes towards this policy was that 28% of the general public thought it promoted inequality, while 17% of service providers shared the same opinion. Additionally, more than half of the general public expressed their preference to increase the number of beneficiaries, as well as the treatment package of the policy, while only around 27% of service providers agreed with this statement.

## 4. Discussion

This study aimed to compare the healthcare priorities of the general public with those of health service providers in Kuwait. Such priorities included how important both groups believe some health services are, their preferred sources for health funding, their attitudes towards the current allocation of health resources, their perceptions of current healthcare costs, and the adequacy of currently available resources. It also aimed to compare both groups’ perceptions regarding two key health policies in the country, namely sending patients abroad for certain types of care and providing private health insurance for retirees. The main finding of this study was that there were some similarities and some differences between health service providers and members of the general public with regard to aspects of priority setting and resource allocation in healthcare and their perceptions regarding some health policies in the country.

Studies evaluating the preferences of members of the general public and health service providers have been performed in several low-, middle-, and high-income countries [20,22,25,29,35]. In a study in Australia, Wiseman found that the preferences of health professionals and members of the general public were similar [38]. Lees et al. [29] alternatively found some differences between the preferences of the public and clinicians regarding health resource allocation.

When respondents were asked to rank some health services, the public and service providers did not have substantial differences in their choices. Both groups’ responses showed a preference for treating the young rather than the old, which was similar to findings from other studies [20,33]. There is a generalised willingness to pay for high-tech lifesaving treatments, which was similar to other studies [33]. In other studies, it was found that the most influential factor in setting priorities was the severity of the disease [22].

Both groups favoured increasing tax on cigarettes as a good option, but it was most popular with service providers, who likely see the effects of smoking on health more directly, while smokers in the general public are less likely to want to pay higher taxes. In a study in the UK, Lees et al. [29] found that 79% of clinicians chose to have a higher tax on cigarettes and alcohol to provide extra money for the NHS. In our study, the implementation of increased income tax as a source of additional funding for health services in the country was exceedingly unpopular. This could be explained by the country’s current situation, which lacks any income tax policy. When compared to the work of Lees et al. [29], they found that 37% of the general public and 55% of clinicians chose to have a higher income tax to increase funding for the NHS in their study. Alternatively, a moderate number (around 40%) of both groups preferred implementing national health insurance as a source of additional funding for health services in the country in this study.

With regard to healthcare resource allocation, the majority of both groups in our study believed that public healthcare should always offer the best possible care, irrespective of costs, similar to the findings of Rosen in Sweden [26].

The age of participants could influence their decisions regarding the allocation of health resources. In a study performed by Werntoft et al. [28] that included individuals who were 65 years of age or above living in southern Sweden, 44% of respondents agreed that patients should pay for their treatment if they have caused their disease themselves. Alternatively, around 69% of the general public and 54% of service providers agreed with the same statement in our study. Another variation was that 32% of older people in Sweden believed that rich people should pay for their treatment [28], whereas the majority of both groups of our study agreed to this statement. More than 85% of respondents from both groups in our study disagreed with the statement “if a disease has an effective treatment, the patient should be treated regardless of the expense,” while 7% of older people in Sweden disagreed with the same statement [28]. 41% of Swedish elderly believed that money was spent on unnecessary things in healthcare [28], whereas around 33% of the general public and 24% of service providers in our study shared the same perception. It was also identified that the age of respondents could influence their perception regarding the adequacy of healthcare resources. Werntoft et al. [28] described that 46% and 49% of respondents from southern Sweden stated that they believed that dental services and elderly care, respectively, had received too few resources. Werntoft et al. [28] showed that there was a demand for increasing the resources allocated to healthcare. Regardless of the categories of care, Table 9 shows that neither the general public nor healthcare providers ever feel that there is “too much” care.

The differences in perception regarding the policy on sending patients abroad for treatment and private health insurance for retirees were very obvious between the two groups included in this study. A possible explanation for this finding might be that one group is better informed than the other group. Otherwise, this could be explained by the composition of the two groups, since Kuwaiti nationals compose the general public group predominantly, whereas Non-Kuwaitis make up the majority of the service providers group.

Previous studies have found various attitudes towards the participation of the public in setting healthcare priorities. Bowling [20] explained that most people in Great Britain wanted to participate in the planning of health services. On the other hand, Litva et al. [40] found differences in the willingness of members of the public to be involved in making healthcare decisions in another study performed in the U.K. Some studies mentioned that the public’s involvement in healthcare rationing depends on the nature of the participation process, and may vary substantially from consultative procedures to delegated citizen power and control [38]. It was argued that public involvement in setting healthcare priorities is vital as it assists in legitimising the process and subsequent outcomes [26,38]. However, some researchers have emphasised the importance of adopting innovative and meaningful methods of incorporating the public’s views in healthcare decision-making [29]. To achieve that, some studies have explained that the public needed to be provided with sufficient information to make healthcare decisions [40]. It was argued that involving the public in healthcare rationing could gradually help to educate people and create a better platform for resource allocation in the future [26]. Rosen [26] debates that starting a dialogue with the public does not implicate the introduction of new claims, but rather establishes support for essential but sometimes unpopular decisions.

Several approaches have been proposed to involve the public in setting healthcare priorities and allocating resources. Rosen [26] explains that information, surveys, and public meetings are various means that could be used in establishing a public dialogue as a first step for involving the public in healthcare rationing. One of the approaches to involving the public in healthcare decision-making was the quality-adjusted life year (QALY)-maximisation model, which was evaluated by Bryan et al. [36]. Another approach to engaging the public in healthcare priority setting is choosing healthplans all together (CHAT). Goold et al. [25] explain that this exercise had several advantages, including ease of use, informativeness, and enjoyment. They continue to describe that the respondents in their study found the information realistic and comprehensive, thought that the group decision-making process was just, and were willing to abide by the group’s decisions [25]. They conclude that CHAT has the potential to be used as a tool for encouraging group discussions, producing collective choices, and including the preferences and values of service users into allocation decisions [25]. Discrete choice experiment (DCE) was another approach for involving the public in healthcare prioritisation [27]. Watson et al. [27] explain that DCE allows for the inclusion of the public’s views in an accessible, transparent, and streamlined decision-making process, which is theoretically valid and practical. On the other hand, Abelson et al. [24] argue that more effective, informed, and meaningful public participation could be achieved using deliberative methods. Mossialos and King [31] explain that procedures such as focus groups, citizens’ juries, and the intensive discussion approach are both informative and deliberative, but require experience in order to be used in public consultation exercises on setting healthcare priorities.

There are several challenges facing public participation in setting healthcare priorities. Evaluating the effect of public participation on policy decision-making remains a difficult task [23]. Litva et al. [40] have found that members of the public understood that their involvement might not lead to change decisions. They add that most of the public had little will to share in the responsibility for healthcare decision-making [40]. Rosen [26] explains that one of the criticisms toward introducing public participation in healthcare rationing was that participants do not usually base their decisions on firm evidence and brotherly feelings. In another study where different responses between doctors, administrators, and members of the public were identified, Rosen and Karlberg [32] argue that such differences in healthcare rationing could be explained by the greater experience and knowledge of doctors and administrators about the costs and benefits of different interventions. They further explain that members of the public might lean towards having an ideal situation, while the practical insight on the unavoidable finiteness of healthcare resources resulted in the more restrictive attitudes of doctors and managers [32]. It was argued that public involvement in setting healthcare priorities should ideally be based on a better public understanding of economic certainties and also on a deeper feeling of responsibility [26,32].

Studies have found that members of the general public reported that the personal experiences and knowledge of the layperson, patients, and their families should complement the expertise of healthcare professionals when making healthcare decisions [38,39]. Wiseman argues that the people who were less educated or who have little knowledge of or experience with healthcare services could find it challenging to answer questions in a practical setting [38]. She then concluded that it is challenging to formulate priority-setting questions that replicate the complexity of real-life decision-making while being easily intelligible to both health professionals and members of the general public [38]. Although some studies have shown that members of the public could choose equal opportunities, fair resource allocation, and equality [22], it was mentioned in other studies that one of the main problems with involving the public in healthcare rationing was that the priorities chosen by the public may not represent the most cost-effective allocation of resources, and may not necessarily offer the most equitable solutions to equal treatment for equal need [20,29]. Some studies mentioned that the public tends to focus on curative services and disregard the more ordinary services, such as mental health services [29,30].

As mentioned in other studies, participants might have felt some restriction in the range and type of their responses because of the use of closed-ended questions [30,38,40]. Additionally, similar to the criticism other studies have received about this method, the ranking of health services question may be considered superficial in relation to the difficulty of the process of setting healthcare priorities, which requires consideration of the cost and effectiveness of treatment and care programmes rather than relying exclusively on values that may include preconceptions [20]. Moreover, this study’s findings allow us to obtain conclusions only related to the situation in Kuwait.

When the public seems to underestimate technical issues, for example, because they do not see daily reminders of the health benefits of diet and exercise, then there should be an educational campaign to inform them. The public may not always understand the cost-effectiveness of procedures, and everybody naturally wants the best possible care. Assuming that healthcare professionals have insights on technical issues that the general public may not, we suggest that large differences indicate where public education through mass media is recommended. This is not to say that the general public is always in the wrong, and, for many issues, our results point to where the government and healthcare providers should be more sensitive to the views of the public. When both the public and the professionals agree, it is the government that should take note.

## 5. Conclusions

This study highlighted the similarities and differences in preferences for health service priority setting between the general public and health service providers in Kuwait. Both groups preferred treating the young rather than the old, while the general public preferred the more expensive health services that had immediate effects rather than health promotion and health services for the elderly. Some differences were predictable, based on the education and experience of healthcare providers, who see the effects of smoking, lack of exercise, and poor diets daily in their work. This indicates the potential for education in shifting public opinion. In any case, differences between health workers and the general public are important to bridge, because the general public may not accept changes in public healthcare if this sector’s problems are not communicated clearly.

Involving the public in setting healthcare priorities is believed to assist in legitimising the process, but it is important to adopt innovative and meaningful methods of incorporating public views in healthcare decision-making. Therefore, the public needs to be provided with sufficient information for them to make healthcare decisions.

## Figures and Tables

**Table 1 healthcare-09-00552-t001:** Government spending on sending patients abroad for treatment in USD for the fiscal years 2012/13, 2013/14, and 2014/15 (Ministry of Finance, 2015).

Entities	Fiscal Year		
	2012/13	2013/14	2014/15
Ministry of Health	379.7 million	393.8 million	1088.0 million
Ministry of Defence	132.3 million	148.0 million	263.9 million
Ministry of Interior	65.6 million	65.6 million	98.5 million
Royal Court	71.2 million	45.3 million	69.9 million
Total	649.2 million	652.8 million	1520.2 million

**Table 2 healthcare-09-00552-t002:** Distribution of healthcare personnel according to the category and type of establishment, Kuwait 2017 [37].

Category	Governmental			Private			Oil			Total		
	K	NK	Total	K	NK	Total	K	NK	Total	K	NK	Total
Doctor	3251	5223	8474	255	2062	2317	55	182	237	3561	7467	11,028
Dentist	1320	559	1879	130	829	959	9	20	29	1459	1408	2867
Nurse	1097	21,606	22,703	42	6751	6793	29	600	629	1168	28,957	30,125
Pharmacist	643	821	1464	331	1252	1583	51	49	100	1025	2122	3147
Others	5523	4306	9829	480	3013	3493	139	99	238	6142	7418	13,560

K: Kuwaiti national, NK: Non-Kuwaiti nationals, Others: lab technicians, radiologists, and physiotherapists.

**Table 3 healthcare-09-00552-t003:** Characteristics of the respondents.

Characteristic	General Public [n (%)]	Health Service Providers [n (%)]
Gender		
Male	595 (49.1)	222 (38.4)
Female	616 (50.9)	356 (61.6)
Age (years)		
21–30	445 (36.8)	109 (18.9)
31–40	420 (34.7)	297 (51.4)
41–50	216 (17.8)	128 (22.2)
51–60	90 (7.4)	32 (5.5)
>60	30 (2.5)	10 (1.7)
No response	10 (0.8)	2 (0.3)
Marital status		
Single	388 (32.0)	106 (18.3)
Married	747 (61.7)	454 (78.6)
Divorced or widowed	61 (5.1)	12 (2.1)
No response	15 (1.2)	6 (1.0)
Nationality		
Kuwaiti	941 (77.7)	210 (36.3)
Non-Kuwaiti	256 (21.1)	366 (63.3)
No response	14 (1.2)	2 (0.4)
Employment status		
Student	28 (2.3)	5 (0.8)
Employed	1044 (86.2)	565 (97.8)
Unemployed	87 (7.2)	6 (1.0)
Retired	41 (3.4)	0 (0.0)
No response	11 (0.9)	2 (0.4)
Monthly Household income (KD)		
<1000	412 (43.0)	279 (48.3)
1000–2000	410 (33.9)	117 (20.2)
2001–3000	163 (13.5)	64 (11.1)
3001–4000	71 (5.9)	31 (5.4)
4001–5000	44 (3.6)	25 (4.3)
>5000	59 (4.8)	54 (9.3)
No response	52 (4.3)	8 (1.4)
Highest degree		
Not completed high school	63 (5.2)	0 (0.0)
High school	118 (9.7)	4 (0.7)
Diploma	357 (29.5)	127 (22.0)
Bachelor’s degree	575 (47.5)	268 (46.3)
Postgraduate degree	86 (7.1)	175 (30.3)
No response	12 (1.0)	4 (0.7)
Governorate of residence		
Capital	367 (30.3)	83 (14.4)
Farwaniya	91 (7.5)	15 (2.6)
Ahmadi	158 (13.1)	233 (40.3)
Jahra	343 (28.3)	119 (20.6)
Hawalli	109 (9.0)	66 (11.4)
Mubarak Al Kabeer	133 (11.0)	46 (8.0)
No response	10 (0.8)	16 (2.7)

**Table 4 healthcare-09-00552-t004:** Respondents’ care-seeking behaviour and their last encounter with the health system.

Question	General Public [n (%)]	Health Service Providers [n (%)]
The last time a healthcare facility was visited		
Less than a month	765 (63.2)	171 (29.6)
One to six months	288 (23.8)	135 (23.4)
Six months to a year	98 (8.1)	137 (23.7)
One to three years	30 (2.5)	58 (10.0)
More than three years	25 (2.0)	66 (11.4)
No response	5 (0.4)	11 (1.9)
Type of healthcare facility last visited		
Public healthcare facility	887 (73.3)	426 (73.7)
Private healthcare facility	299 (24.7)	126 (21.7)
Healthcare facility overseas	16 (1.3)	13 (2.3)
No response	9 (0.7)	13 (2.3)
Method of payment for healthcare Private health insurance Out-of-pocket payment Use public health services (free of charge) Use a combination of methods No response	229 (18.9) 254 (21.0) 417 (34.4) 299 (24.7) 12 (1.0)	29 (5.0) 165 (28.6) 247 (42.7) 120 (20.8) 17 (2.9)

**Table 5 healthcare-09-00552-t005:** Mean priority ranking of health services (1 = highest priority).

Health Services	General Public		Service Providers	
	Mean	Rank	Mean	Rank
Treatments for children with life threatening illnesses (i.e., leukaemia)	2.13	1	2.17	1
Special care and pain relief for people who are dying (i.e., untreatable cancer)	3.19	2	4.66	4
High technology surgery, organ transplants and procedures which treat life threatening conditions	4.33	3	4.64	3
Preventive screening services and immunization	5.19	4	3.43	2
Surgery, such as knee replacement, to help people carry out everyday tasks	5.44	5	5.25	7
Intensive care for premature babies with only a slight chance of survival	5.67	6	5.09	6
Treatment for people aged 75 and over with life threatening illness	6.01	7	6.67	10
Psychiatric services for people with mental Illness	6.28	8	5.44	8
Treatment for infertility	6.47	9	6.84	11
Health promotion/education services to help people lead healthy lives	6.63	10	4.91	5
Long stay hospital care for elderly people	7.16	11	7.55	12
Nursing and community services at home	7.56	12	6.38	9

**Table 6 healthcare-09-00552-t006:** Preferred sources of additional funding for healthcare services in Kuwait.

Sources of Additional Funding for Healthcare Service	General Public [n (%)]	Service Providers [n (%)]
Higher tax on cigarettes *	711 (58.7)	420 (72.7)
Implementation of national health insurance	494 (40.8)	212 (36.7)
Tax on pollution (i.e., cars and factories)	459 (37.9)	216 (37.4)
Decrease the budget allocated for sending patients abroad for treatment *	230 (19.0)	257 (44.5)
Decrease the budget of other Ministries such as the Ministry of Defence	211 (17.4)	85 (14.7)
Implementation of user charges for public healthcare services	125 (10.3)	59 (10.2)
Implementation of income tax	69 (5.7)	46 (8.0)
Other	137 (11.3)	48 (8.3)

* The difference between the percentage of the general public and the service providers selecting this option is statistically significant (*p* < 0.05).

**Table 7 healthcare-09-00552-t007:** Attitudes about the allocation of healthcare resources in Kuwait.

Question	General Public [n ‘yes’ (%)]	Service Providers [n ‘yes’ (%)]
Should public healthcare always offer the best possible care, irrespective of costs? *	1021 (84.3)	373 (64.5)
Should we invest more public resources in public Healthcare?	975 (80.5)	476 (82.4)
Should decision-makers in the Ministry of Health make more explicit prioritisations? *	972 (80.3)	330 (57.1)
Should the general public be offered more opportunities to influence healthcare resource allocation? *	678 (56.0)	432 (74.7)
Should simpler treatments or healthcare services be paid by the patients themselves (i.e., dental scaling)?	383 (31.6)	187 (32.4)
Do you think that public healthcare resources are sufficient to satisfy all healthcare need? *	321 (26.5)	230 (39.8)
Do you think that public healthcare resources are sufficient to always offer patients best possible care? *	301 (24.9)	246 (42.6)
Do you think that decision-makers in the Ministry of Health handle prioritisations in a good manner? *	157 (13.0)	185 (32.0)

* The difference between the percentage of the general public and the service providers who responded with ‘yes’ to the questions is statistically significant (*p* < 0.05).

**Table 8 healthcare-09-00552-t008:** Responses to statements on healthcare costs in Kuwait.

Question		General Public [n (%)]	Service Providers [n (%)]
If a disease has effective treatment, the patient should be treated regardless of the expense	Agree	75 (6.2)	44 (7.6)
	No opinion	81 (6.7)	41 (7.1)
	Disagree	1055 (87.1)	493 (85.3)
If two types of treatment exists, the cheaper one should be chosen, even if it is less effective *	Agree	947 (78.2)	499 (86.3)
	No opinion	103 (8.5)	40 (6.9)
	Disagree	161 (13.3)	39 (6.8)
Money is spent on unnecessary things in healthcare *	Agree	394 (32.5)	138 (23.9)
	No opinion	335 (27.7)	118 (20.4)
	Disagree	482 (39.8)	322 (55.7)

* The difference between the percentage of the general public group and the service providers’ group’s responses is statistically significant (*p* < 0.05).

**Table 9 healthcare-09-00552-t009:** Respondents’ perception on the adequacy of resource allocation.

Health Service		General Public [n (%)]	Service Providers [n (%)]
Psychiatric care *	Too little	506 (41.8)	150 (26.0)
	Enough	176 (14.5)	150 (26.0)
	Too much	24 (2.0)	6 (1.0)
Health education and prevention	Too little	531 (43.9)	256 (44.3)
	Enough	424 (35.0)	211 (36.5)
	Too much	44 (3.6)	13 (2.3)
Dental services *	Too little	367 (30.3)	108 (18.7)
	Enough	616 (50.9)	306 (52.9)
	Too much	75 (6.2)	42 (7.3)
Elderly care *	Too little	444 (36.7)	168 (29.1)
	Enough	440 (36.3)	206 (35.6)
	Too much	67 (5.5)	58 (10.0)
Hospital care *	Too little	608 (50.2)	179 (31.0)
	Enough	403 (33.3)	281 (48.6)
	Too much	49 (4.1)	45 (7.8)
Primary healthcare *	Too little	454 (37.5)	162 (28.0)
	Enough	452 (37.3)	289 (50.0)
	Too much	38 (3.1)	35 (6.1)
End-of-life care *	Too little	473 (39.1)	150 (26.0)
	Enough	243 (20.1)	205 (35.5)
	Too much	49 (4.1)	29 (5.0)
Drug addiction/rehabilitation care *	Too little	428 (35.3)	151 (26.1)
	Enough	208 (17.2)	158 (27.3)
	Too much	51 (4.2)	8 (1.4)
Healthcare information	Too little	510 (42.1)	261 (45.2)
	Enough	324 (26.8)	194 (33.6)
	Too much	41 (3.4)	19 (3.3)
Healthcare administration *	Too little	489 (40.4)	176 (30.5)
	Enough	309 (25.5)	205 (35.5)
	Too much	47 (3.9)	47 (8.1)
Child care *	Too little	451 (37.2)	150 (26.0)
	Enough	432 (35.7)	283 (49.0)
	Too much	61 (5.0)	24 (4.2)

* The difference between the percentage of the general public group and the service providers’ group’s responses is statistically significant (*p* < 0.05).

**Table 10 healthcare-09-00552-t010:** Respondents’ opinions on the policy on sending patients abroad for treatment.

Question	Responses	General Public [n (%)]	Service Providers [n (%)]
The policy on sending patients abroad is costly/expensive *	SA and A	555 (59.0%)	111 (52.9%)
	N	334 (35.5%)	95 (45.2%)
	D and SD	52 (5.5%)	4 (2.0%)
The policy on sending patients abroad is misused *	SA and A	586 (62.3%)	122 (58.1%)
	N	293 (31.1%)	82 (39%)
	D and SD	62 (6.6%)	6 (2.9%)
The policy on sending patients abroad is politically driven *	SA and A	528 (56.1%)	106 (50.5%)
	N	388 (41.2%)	100 (47.5%)
	D and SD	25 (2.7%)	4 (2.0%)
Most cases sent abroad for treatment without real medical indication *	SA and A	475 (50.5%)	89 (42.4%)
	N	369 (39.2%)	104 (49.5%)
	D and SD	97 (10.3%)	17 (8.1%)
Most specialised treatments are available locally *	SA and A	303 (32.2%)	80 (38.1%)
	N	454 (48.2%)	102 (48.5%)
	D and SD	184 (19.6%)	28 (13.4%)
It is a constitutional right to have the option of being sent abroad for treatment *	SA and A	413 (43.9%)	64 (30.4%)
	N	460 (48.9%)	106 (50.5%)
	D and SD	68 (7.2%)	40 (19.1%)
Sending patients abroad for treatment has several advantages *	SA and A	616 (65.5%)	90 (42.9%)
	N	301 (32.0%)	106 (50.5%)
	D and SD	24 (2.5%)	14 (6.6%)
Sending patients abroad for treatment has decreased the trust in the local health system *	SA and A	475 (50.4%)	109 (51.9%)
	N	378 (40.3%)	93 (44.2%)
	D and SD	88 (9.3%)	8 (3.9%)
Patients prefer to be treated in their home country around their families	SA and A	478 (50.8%)	65 (31.0%)
	N	375 (39.9%)	112 (53.3%)
	D and SD	88 (9.3%)	33 (15.7%)

SA and A = strongly agree and agree, respectively; N = neither agree nor disagree; D and SD = disagree and strongly disagree, respectively. * The difference between the percentage of the general public group and the service providers’ group’s responses is statistically significant (*p* < 0.05).

**Table 11 healthcare-09-00552-t011:** Respondents’ opinions on the policy on private health insurance for retirees (Afya).

Question	Responses	General Public [n (%)]	Service Providers [n (%)]
The health insurance for retirees policy is a good policy *	SA and A	601 (63.9%)	71 (33.9%)
	N	296 (31.4%)	116 (55.2%)
	D and SD	44 (4.7%)	23 (10.9%)
The health insurance for retirees policy has clear objectives *	SA and A	479 (50.9%)	57 (27.1%)
	N	380 (40.4%)	107 (51%)
	D and SD	82 (8.7%)	46 (21.9%)
The health insurance for retirees policy decreased load on public health services *	SA and A	531 (56.4%)	93 (44.3%)
	N	366 (38.9%)	101 (48.1%)
	D and SD	44 (4.7%)	16 (7.6%)
The health insurance for retirees policy promoted patient choice *	SA and A	608 (64.6%)	93 (44.3%)
	N	314 (33.4%)	109 (51.9%)
	D and SD	19 (2.0%)	8 (3.8%)
The health insurance for retirees policy is misused	SA and A	122 (13.0%)	22 (10.5%)
	N	540 (57.4%)	120 (57.1%)
	D and SD	279 (29.6%)	68 (32.4%)
The health insurance for retirees policy is a step towards privatising healthcare *	SA and A	42 (4.5%)	11 (5.3%)
	N	451 (47.9%)	124 (59.0%)
	D and SD	448 (47.6%)	75 (35.7%)
The health insurance for retirees policy promotes inequality *	SA and A	266 (28.3%)	35 (16.6%)
	N	458 (48.6%)	132 (62.9%)
	D and SD	217 (23.1%)	43 (20.5%)
The beneficiaries of the health insurance for retirees policy should be increased *	SA and A	509 (54.1%)	57 (27.2%)
	N	373 (39.6%)	120 (57.1%)
	D and SD	59 (6.3%)	33 (15.7%)
The treatment package of the health insurance for retirees policy should be increased *	SA and A	563 (59.8%)	58 (27.7%)
	N	355 (37.7%)	121 (57.6%)
	D and SD	23 (2.5%)	31 (14.7%)

SA and A = strongly agree and agree, respectively; N = neither agree nor disagree; D and SD = disagree and strongly disagree, respectively. * The difference between the percentage of the general public group and the service providers’ group’s responses is statistically significant (*p* < 0.05).

## Data Availability

The data is not publicly available. It was treated with confidentiality and was only accessible by the researchers.

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
