# Peer review of "Comparing Public and Provider Preferences for Setting Healthcare Priorities: Evidence from Kuwait"

_healthcare, 2021, doi:10.3390/healthcare9050552_

Round 1

Reviewer 1 Report

Comparing public and provider preferences for setting healthcare priorities: Evidence from Kuwait

This study aims to compare the priorities of the public with those of health service providers in Kuwait. With the objective to compare the perceptions of providers and beneficiaries of healthcare regarding key health policies, the authors made a questionnaire asking to rank health services by their perceived importance; rank preferred sources of additional health funding; and share their perceptions of the current allocation of health resources. To analyze the data, the authors used descriptive analytical methods were used to analyse the data. The main results suggest that the general public may not accept changes in public healthcare if this sector's problems were not communicated clearly. Therefore, the public needs to be provided with sufficient information for them to make healthcare decisions.

This manuscript analysis an interesting topic to the research community and the public in general. It is well structured and written. I have a minor comment, I think the abstract is too long.

Author Response

Dear Mam/Sir,

I hope this finds you well. Thank you for your comments. I have shortened the abstract in the revised version.

Reviewer 2 Report

Short summary

The research aims to compare the priorities and the perceptions on key health policies of the public with those of health service providers in Kuwait. In fact, the Kuwaiti government may no longer have sufficient funds to finance all previously funded health services and may need to engage in an explicit priority setting process to improve the efficiency of the system. The analysis is based on a survey over a sample of health service providers and the general public in Kuwait. The main finding of this study is that there were some similarities and some differences between health service providers and members of the general public with regards to aspects of priority setting and resource allocation in healthcare and their perceptions regarding some health policies in the country.  Both groups preferred treating the young rather than the old, while the general public preferred the more expensive health services that had immediate effects rather than health promotion and health services for the elderly. Such differences might suggest that the general public may not accept changes in public healthcare if this sector's problems were not communicated clearly. Involving the public in setting healthcare priorities is believed to assist in legitimising the process, but it is important to adopt innovative and meaningful methods of incorporating public views in healthcare decision-making. Therefore, the public needs to be provided with sufficient information for them to make healthcare decisions.

Broad comments

The manuscript illustrates a research that seems to have solid statistical basis, however the analysis lacks of graphical representations illustrating succintly the many arguments covered while presenting results and discussing the major findings. Furthermore, more political issues concerning, as an example, the need of consultations to integrate policy makers' decisions and a focus on regulation impact analysis may enrich the sections "discussion" and "conclusions". 

Specific comments

Tables 4-12 could be placed in an ad hoc appendix, while in the main body  they may be substituted by graphs (as bar charts) representing at least the most important data commented.

Author Response

Dear Mam/Sir,

I hope this finds you well and thank you for invaluable comments. Please find below our responses regarding your comments.

Reviewer wants tables in the appendix, and represented by graphs in the text.

AMS: We have considered whether to make a graphic of each table separately, based on the best way to present the material. Table 4 is best presented as a table, as there are so many categories, we think that a graph would be more difficult to understand. We find this to be true of most tables. We interpret the comment as an indication that the tables could be presented in an easier to process form, and have reordered many of them to make a comparison easier.

Awaiting your kind response.

Best wishes,

Abdullah

Reviewer 3 Report

General comment

First of all, I congratulate the authors on the subject matter. I consider it of great importance for the area of health planning. 

In general, the authors have a general objective, which is answered in the manuscript. The ideas are clear, the method appropriate and very well described, presentation of the data clear, the discussion could have an improved form of presentation and the conclusion also.

Abstract

1. The abstract is too long. It is suggested that in the presentation of the results, a summary of what are the differences and similarities of the opinions of the public and health providers should be made. It is suggested to look again at the results and concisely select those that are the main findings. 

Introduction

1. Insert the source of Table 1.
2. It is suggested to add at the end of the introduction a paragraph with the structure of the manuscript. It may help the reader. 
3. Review formatting, there are letters with different sizes.

Methodology

1. In Table 3 it is suggested that "other" be described in the table. The values are high to be grouped under "other". 
2. It is suggested to explain in a note in table 3 what is meant by health providers linked to "oil". 
3. In the first paragraph of the subsection "data collection" we suggest removing the mention of qualitative questions, because they were not used in this manuscript.
4. We suggest to revise this subsection because there is repeated information in different paragraphs, e.g. survey applied is Arabic and English; e.g. survey conducted in February 2018. 

Results

1. In the explanation of table 6 it is suggested that it is highlighted that the public and healthcare providers, out of 12 priorities, agree with 2 out of the top 3 priorities. 
2. In table 7 insert a note explaining what "others" are.
3. In the explanation of table 7 suggest mentioning the second option of the general public. And highlighting the third which is a coincident option for both groups. 

4. In the explanation of table 8 there is repeated information about health providers being convinced that the general public should have a greater opportunity to influence the allocation of health care resources. It is suggested to start the sentence at "Around 75%..."
5. Table 8 seems to indicate that the population trusts governments' ability to prioritise less than healthcare providers, and they point to wanting more investment. This idea should be discussed in the "discussion" section. 
6. In the explanation of table 10 make it more evident that the "too much" option did not achieve a majority for any service/assertive. And this can try to be debated in the discussion section. 
7. Table 12 should appear right after its mention in the text. 
8. On the insurance policy for retirees there is more divergence between the two groups. It is suggested that this result should be made more explicit. 
9. Has any analysis been carried out comparing the views of separate health care providers (e.g. public, private)? And between these and the general public? Although the aim of the manuscript is to compare the views of the two groups only (general public and health care providers), there may be large differences in views between healthcare providers, particularly because the paper only assesses the public health service. Is it possible to do this analysis? Has any analysis been carried out in this respect? 

Discussion
The discussion is interesting and the authors seek to compare the results with other findings in other investigations. In addition, it resumes the discussion on the advantages and challenges of population participation in health priority setting, supported by the literature.
However, the discussion ends up repeating many of the results presented in the "Results" section, presenting percentages already known to the reader. Thus, the suggestion is that the authors try to elaborate a matrix/table with the global positions that were evaluated, indicating those in which there is convergence and divergence between the two groups. In this case, it would avoid the repetition of the percentages, which are already very well described in the "Results" section. I consider that the manuscript will gain from this synthesis of the results. 

Conclusion

The conclusion is shallow for the richness of the results presented, although the last paragraph is good. The suggestion is that on the basis of the suggested table/matrix, the authors might actually reflect on what are the structural similarities and divergences of the views of the two groups studied. In the way it is presented, it seems that the authors conclude or make an assertion based on only two results. I believe that the conclusion should be further enriched. 

Author Response

Dear Mam/Sir,

I hope this finds you well and thank you for your invaluable comments. In the following lines, I will address all the comments.

Reviewer: “The abstract is too long. It is suggested that in the presentation of the results, a summary of what are the differences and similarities of the opinions of the public and health providers should be made. It is suggested to look again at the results and concisely select those that are the main findings”

AMS: The abstract has been shortened. We have presented the most important aspects of our findings, and trimmed unnecessary details.

Reviewer: “Review formatting, there are letters with different sizes.”

AMS: We found one instance of inconsistent font in the introduction and a few in the remaining body of the article, and changed it.

Reviewer: “Insert the source of Table 1.”

AMS: The source was inserted between brackets at the end of the caption. It was from the Ministry of Finance in the year 2015.

Reviewer: “...Add a paragraph with the structure of the manuscript. It may help the reader.” 

AMS: We disagree with this suggestion. The reader does not need a summary of the paper’s structure, as it is clear to see from the headings and subheadings of each section. The structure is standard and expected for an academic paper. We have seen this convention occasionally, and think that it is a spill-over from verbal presentations, where the audience needs an idea of the structure of a talk to keep them engaged, because they do not have a document to scan ahead.

Reviewer: “...explain in a note in table 3 what is meant by health providers linked to "oil".”

AMS: A sentence has been added (line 166) that describes this heading.

Reviewer: “ "other"should be described in the table. The values are high to be grouped under "other". 

AMS: “Others” are specified below the table (“Others: lab technicians, radiologists, and pharmacists”. Unfortunately, we can't do much in dividing this group since it was mentioned as it is in the reference (MoH report).

Reviewer: “In the first paragraph of the subsection "data collection" we suggest removing the mention of qualitative questions, because they were not used in this manuscript.”

AMS: Qualitative questions were used. The reviewer may have been misled by our characterization of the questions coming from a “forthcoming paper”. We have clarified the wording on this.

Reviewer: “In the first paragraph of the subsection "data collection" we suggest removing the mention of qualitative questions, because they were not used in this manuscript.”

AMS: We agree that it was confusing when we read it after receiving the reviewers comments. We have developed some questions by going back to a qualitative study that we have conducted and was already published. We have clarified the wording on this (lines 177, 178 and 179).

Reviewer: “... revise this subsection because there is repeated information in different paragraphs, e.g. survey applied is Arabic and English; e.g. survey conducted in February 2018.”

AMS: We have removed these redundancies.

Reviewer: “ In the explanation of table 6 ... highlight that the public and healthcare providers, out of 12 priorities, agree with 2 out of the top 3 priorities.“

AMS: A sentence was added to emphasize this. The table was also modified so that priorities are in rank order of the general public. This makes comparisons easier between the two groups, and also makes a table a better format for this information than a graph.

Reviewer: “suggest mentioning the second option of the general public. And highlighting the third which is a coincident option for both groups.”

AMS: We disagree that the content that was clearly presented in a table should be repeated in the text. However, to make it more clear, we have reordered the content based on that general public ranking, as was done in the previous table.

Reviewer: insert a note explaining what "others" are.

AMS: The explanation is described starting in line 319.

Reviewer: “In the explanation of table 8 there is repeated information about health providers being convinced that the general public should have a greater opportunity to influence the allocation of health care resources. It is suggested to start the sentence at "Around 75%…"”

AMS: We agree and now start the text as suggested. We also reordered the table as in the previous two, for clarity.

Reviewer: “In the explanation of table 10 make it more evident that the "too much" option did not achieve a majority for any service/assertive. And this can try to be debated in the discussion section.”

AMS: Good point. We have started our discussion with a new sentence on this point.

Reviewer: “Table 12 should appear right after its mention in the text.”

AMS: Done, thanks.

Reviewer: “Has any analysis been carried out comparing the views of separate health care providers (e.g. public, private)? And between these and the general public? Although the aim of the manuscript is to compare the views of the two groups only (general public and health care providers), there may be large differences in views between healthcare providers, particularly because the paper only assesses the public health service. Is it possible to do this analysis? Has any analysis been carried out in this respect?”

AMS: These analyses have not been done, and while we appreciate the suggestion for future work, we think that reviewers should not expand a paper beyond the author’s intentions.

Reviewer: “On the insurance policy for retirees there is more divergence between the two groups. It is suggested that this result should be made more explicit.”

AMS: As we stated, differences “very obvious between the two groups included in this study.”  We find this to be explicit.

Reviewer: “Table 8 seems to indicate that the population trusts governments' ability to prioritise less than healthcare providers, and they point to wanting more investment. This idea should be discussed in the "discussion" section.”

AMS: We disagree with the reviewer’s interpretation of Table 8, and find the table to be clear.

Reviewer: “In  table 10  the "too much" option did not achieve a majority for any service/assertive.  this could be debated in the discussion.”

AMS: Sentence added to the end of the paragraph (line 157).

Reviewer: “The conclusion is shallow for the richness of the results presented, although the last paragraph is good. The suggestion is that on the basis of the suggested table/matrix, the authors might actually reflect on what are the structural similarities and divergences of the views of the two groups studied. In the way it is presented, it seems that the authors conclude or make an assertion based on only two results. I believe that the conclusion should be further enriched.”

AMS: We have added content to the conclusions to generalize the importance of our findings.

Finally, we were grateful for the comments and hope that our responses were satisfactory for the editorial team. Hopefully, our work would be published in your journal.

Regards,

Abdullah Alsabah

Round 2

Reviewer 3 Report

Dear Author(s),
I hope this e-mail finds you well.

I am pleased with the author's answers. Despite some points, on which we disagree, there is no harm to the idea/structure of the manuscript. 
The manuscript is relevant and of great interest. 

Best regards,
Katielle Silva

Author Response

Dear Katielle Silva,

I hope this finds you well and thank you so much for your invaluable comments.

Best wishes,

Abdullah Alsabah